# Evaluation of a Dog Population Management Intervention: Measuring Indicators of Impact

**DOI:** 10.3390/ani10061061

**Published:** 2020-06-19

**Authors:** Gemma C Ma, Ann-Margret Withers, Jessica Spencer, Jacqueline M Norris, Michael P Ward

**Affiliations:** 1Sydney School of Veterinary Science, Faculty of Science, The University of Sydney, Camperdown 2006, Australia; jacqui.norris@sydney.edu.au (J.M.N.); michael.ward@sydney.edu.au (M.P.W.); 2RSPCA NSW, Yagoona 2199, Australia; amwithers@rspcansw.org.au; 3Aboriginal Environmental Health Unit, NSW Health, Dubbo 2830, Australia; Jessica.Spencer@health.nsw.gov.au

**Keywords:** dog, dog population management, monitoring, evaluation, impact, euthanasia, dog bite

## Abstract

**Simple Summary:**

Dogs are important companions to people but can also present challenges to the health and safety of communities if their populations are not effectively managed. Poorly managed dog populations can increase the risk of dog bites to people and other animals, spread disease and create a public nuisance. Dog overpopulation can also lead to compromised dog welfare. Dog population management (DPM) interventions are necessary in communities where veterinary services are not available or are not used by dog owners, or where there are large numbers of free-roaming unowned dogs. Ongoing monitoring and evaluation of DPM interventions is necessary to ensure they are effective. We evaluated a DPM intervention using readily available data collected from the intervention itself and publicly available data from secondary sources. We show the intervention resulted in significant increases in the proportion of dogs with permanent identification (microchips) and that had been neutered. We also show the intervention was associated with significant reductions in dog attacks and euthanasia of dogs in participating communities across at least three years. The decrease in euthanasia was associated with an increase in the release of dogs by council pounds to other organizations for rehoming. We present additional useful indicators to measure community engagement with caring for dogs and suggest benchmark figures to guide planning and monitoring of similar interventions in future.

**Abstract:**

Dogs are important companions to people but can also present challenges to health and safety of communities if their populations are not effectively managed. Dog population management (DPM) is often undertaken by individual dog owners; however, some communities require additional DPM interventions, especially when veterinary services are unavailable or underutilised. This study evaluated the effectiveness of a DPM intervention conducted in 13 communities between September 2016 and November 2019 and assessed the utility of routinely collected data—program metrics and secondary data collected by local governments—to measure indicators of impacts. The intervention resulted in significant increases in the proportion of dogs presenting that were microchipped and surgically sterilised in participating communities. The intervention also resulted in significant reductions in dog attack incidents and euthanasia of dogs in council pounds in communities that participated for three or more years. Ongoing monitoring and evaluation of DPM interventions is critical to determine if impacts are being achieved. This study demonstrates the potential benefits of a DPM intervention for community safety and dog welfare and highlights the utility of routinely collected data. We also suggest benchmarks for indicators of community engagement to guide planning and monitoring of similar interventions.

## 1. Introduction

Dogs are valued companions of people and an important component of many human societies worldwide. However, dogs can also be a hazard to human health and community safety. Dog bites can cause serious injuries and psychological trauma, especially in children [1]. Dogs can also transmit zoonotic diseases and parasites and can contribute to poor household and community environmental health [2]. Rabies does not occur in dogs in Australia. However, in many parts of the world, including Australia’s nearest neighbours, rabies is a major motivator for effective dog population management (DPM). Rabies causes an estimated 59,000 global human deaths per year, 99% of which are the result of a dog bite [3,4,5]. Effective DPM is important to ensure adequate animal welfare standards are maintained, pathogen transmission among dogs and to humans is minimised and dog attacks on people and other animals are avoided [6]. Poor management of dog populations also puts capacity and resource strain on council pounds and shelters, resulting in substantial financial cost to communities and challenges to animal welfare [7]. Effective DPM—especially in remote communities of the Top End—is also an important component of Australia’s disease preparedness measures in the event of a rabies incursion from neighbouring Indonesia [8]. 

Dog population management aims to maintain the size, composition, and stability of dog populations, especially by reducing unplanned breeding, increasing longevity of individual dogs and decreasing population turnover [6,9]. In many situations, DPM is mostly conducted by individual dog owners. This is the legislative expectation in Australia, where a moral imperative to be “responsible dog owners” is placed on individuals [8]. However, additional interventions are often required to ensure community safety, especially when veterinary services are unavailable, inaccessible, or unaffordable such as in remote regions and in disadvantaged or marginalised communities [10,11,12]. Globally, interventions to manage dog populations often focus on reproductive control using surgical sterilisation (i.e., permanent, surgical contraception; otherwise referred to as ‘desexing’ or ‘neutering’) or chemical contraception variably coupled with other components such as vaccination (including for rabies in trap-neuter-vaccinate-release programs), permanent identification, registration of individual dogs, provision of veterinary treatment and improving avenues for holding and rehoming dogs [9,13]. 

Monitoring and evaluation of DPM interventions—particularly their impacts on achieving community engagement, community safety and dog population turnover—remains an important challenge [13]. Evidence of the impact of these interventions is limited, especially in Australia. Evidence from long-running Animal Birth Control programs in India have demonstrated positive impacts on dog population size and demographics [14,15], dog health and welfare [16,17] and human dog bite incidence [18]. However, the methods used to measure impacts in such studies are rarely useful for ongoing intervention evaluation. Dog censuses—where each dog in a target population is counted—or mark re-sight methodologies—where target population demographics are extrapolated from the number and characteristics of dogs seen in street surveys—have been recommended [6,15]. However, these methods can be expensive, labour intensive and logistically difficult and will not effectively measure impacts such as community engagement and community safety.

This study used a natural experiment to evaluate the effectiveness of a DPM intervention to achieve its objectives of improving community safety, improving animal welfare, and increasing community engagement with dog health activities. We also assessed the utility of routinely collected and readily available secondary source data in ongoing monitoring and evaluation of DPM interventions.

## 2. Materials and Methods 

### 2.1. Study Context

This study evaluated a DPM intervention, the Indigenous Community Companion Animal Health Program (ICCAHP) conducted by the Royal Society for the Prevention of Cruelty to Animals (RSPCA) New South Wales (NSW) in communities across NSW between September 2016 and November 2019. The ICCAHP aimed to address the companion animal health service gap that exists in many Aboriginal communities.

Communities were identified for participation in the program based on the following criteria: Aboriginal community-identified need for a dog health intervention; a large proportion of the population who identify as Aboriginal; and socioeconomic disadvantage based on a ranking under the Australian Bureau of Statistics (ABS) Index of Relative Socio-Economic Advantage and Disadvantage in the lowest quartile for the state [19]. Participating communities varied substantially from large inner regional centres to small, remote communities; from coastal regions to the arid inland (Table 1). Some were discrete Aboriginal communities, previously Aboriginal reserves or missions, other Aboriginal communities were part of rural and remote townships. Geographical proximity to a veterinarian also varied. However, our previous research demonstrates that barriers to accessing veterinary services, particularly cost and access to transport, result in low levels of engagement between Aboriginal communities and local veterinary services regardless of their proximity [12]. Rates of dog ownership in Aboriginal communities in Australia are high, with around 65% of households owning at least one dog [20]. Dogs in the participating communities are mainly kept for companionship and for the protection of person and property [12]. They are highly valued, considered family members and often allowed indoors, including sharing beds with their human family [12]. The DPM history varied substantially between study communities. All had considerable gaps in access to veterinary services in preceding years. Some had received similar programs from RSPCA NSW previously for example in 2014 in Community 5 and Community 8 and in 2009 in Community 4 and Community 6. In all participating communities most of the DPM challenges are believed to relate to dogs that are owned but allowed to roam with smaller contributions from unowned free-roaming dogs.

### 2.2. The Intervention: RSPCA’s ICCAHP

The ICCAHP was designed in consultation with the communities that initially invited the intervention in 2016 and was tailored to each individual community. Funding for ICCAHPs was provided by NSW Health, Aboriginal Environmental Health Unit and planning and delivery of ICCAHPs was conducted jointly between RSPCA NSW and Aboriginal Environmental Health Project Officers. The intervention involved visits to communities by a multi-disciplinary team consisting of veterinarians, veterinary nurses and Community Outreach Officers to provide basic veterinary services for dogs and cats. Only dogs were included in this study. Effective cat population management faces very different challenges to effective DPM; hence, we consider it appropriate to examine these species separately. 

Timing, duration and location of ICCAHPs in each community was arranged in consultation with local Aboriginal community governance bodies. These organisations assisted with local promotion of the program in the weeks prior to the ICCAHP and assisted with recruiting local community members to work with RSPCA NSW teams as paid community liaison officers during the ICCAHP. Local government (council) representatives, particularly animal control officers, were encouraged to participate in ICCAHPs as an opportunity to build positive relationships with their community, for example by providing microchips, free life-time registration of dogs, or by assisting participants to transport their animals to veterinary visits during and after the ICCAHP.

Equipment required to create a registered veterinary field hospital was transported directly to geographically remote communities allowing surgical sterilisation to be performed on-site during the program in communities where no veterinary clinic was present. Alternatively, in communities with a local veterinary clinic, surgical sterilisation was scheduled and paid for by the ICCAHP through local veterinarians. Eligible participants—those residing within the township and eligible for federal government assistance (presenting with either a health care card or pension card) accessed free veterinary consultations, vaccination (canine distemper virus, canine adenovirus type 1 and canine parvovirus), permanent identification (microchip implantation), internal and external parasite treatments and treatment of sick or injured dogs. Participants were also able to have unwanted or suffering dogs humanely euthanased on request or transported to a rescue organisation for rehoming if suitable. The ICCAHP was preceded by visits to local schools by RSPCA Education Officers and NSW Health, Aboriginal Environmental Health Officers where content on dog safety and basic dog care was presented and linked to human health (for example, hand hygiene).

### 2.3. Indicators Measured

This study adapted the evaluation framework proposed by International Companion Animal Management Coalition (ICAM) [6], using proxy indicators to measure impacts (Table 2).

#### 2.3.1. ICCAHP Indicators

Data on all dogs participating in ICCAHPs between September 2016 and November 2019 were collected. This included demographic information (age, sex), whether the dog had already been surgically sterilised or was entire, whether the dog had a microchip, whether the dog received microchipping or surgical sterilisation during the ICCAHP and whether the dog was a return participant from an ICCAHP in a previous year.

#### 2.3.2. Secondary Source Indicators

Council pound statistics and council records of dog attack incidents were obtained at the Local Government Area (LGA) level from the NSW Office of Local Government for the period 1 July 2012 to 30 June 2019. In this instance, a “council pound” under the NSW Companion Animals Act 1998 means any public or private pound established or approved by a council as a place for the holding of animals for the purposes of the Act [22]. In other jurisdictions, these may be referred to as municipal, public or government-funded animal shelters. Council pound data analysed included the total number of dogs entering the pound during the financial year, the number of dogs reclaimed by their owners and the total number of dogs euthanased. Reasons for euthanasia included illness/disease/injury, feral/infant, owner request, unable to rehome, unsuitable for rehoming and dangerous/restricted/other. Dog attack incidents were defined, according to the NSW Companion Animals Act 1998, as any incident in which a dog rushes at, attacks, bites, harasses or chases any person or animal (other than vermin), whether or not any injury is caused to the person or animal. Secondary source indicators were standardised based on the estimated resident population in the respective LGA in each year to allow comparisons between LGAs and with total values for the whole of NSW and regional NSW [23].

Dog attack incidents and council pound statistics—including total number of dogs admitted, number of dogs reclaimed by their owners and total number of dogs euthanased—were reported for the LGA of each community receiving an ICCAHP in the study period. Note that some LGAs contained more than one ICCAHP community (Table 1).

Council amalgamations occurred in NSW in May 2016 affecting 39 of the state’s previous 152 LGAs, creating 17 new LGAs (a total of 130 new LGAs). One community in the present study was affected by the council amalgamations (Community 1). These changes to LGAs had some effects on council pound statistics due to pound closures and mergers. 

#### 2.3.3. Qualitative Participant Evaluation

Semi-structured interviews were conducted with participants in ICCAHPs between September 2018 and November 2019 as previously described [12]. ICCAHP participants were asked about positive and negative aspects of their experience with the ICCAHP and how they thought the ICCAHP could be improved. If respondents had participated in an ICCAHP in a previous year, they were also asked if they noticed any changes in their pet and to describe these changes if applicable.

### 2.4. Analyses

Statistical analyses were conducted using SPSS^®^ Statistics Version 24 (IBM^®^, Armonk, NY, USA). Associations between presentation at the ICCAHP and sex, proportion already microchipped and proportion already sterilised were tested using the Pearson Chi-square test. Mean age of dogs was compared using an Independent-Samples Kruskal-Wallis Test. Before and after ICCAHP implementation, an analysis was conducted on pooled data of key metrics (dog attack incidence, total number of dogs admitted to council pound, total number of dogs euthanased, dogs released to organization for rehoming and dogs released to owners, all per 1000 population) using ANOVA with a factorial block design (LGA as block factor) and a Tukey HSD All-Pairwise Comparison Test. Assumptions of normality were tested using the Shapiro-Wilk Normality Test.

### 2.5. Ethics

Ethical approval was granted by the University of Sydney Human Research Ethics Committee (Project number 2018/763).

## 3. Results

### 3.1. ICCAHP Primary Indicators

In total 1633 dogs, belonging to 916 households, were presented during 26 ICCAHPs in 13 communities (Table 1). Most dogs presenting to the program were crossbreds (1125 of 1633, 69%). The age distribution of dogs presented was positively skewed with mean age 1.5 years, median 2.9 years and range of eight weeks to 20 years (Figure 1a). The number of male and female dogs presented to ICCAHPs was not significantly different (53% male, 46% female, 0.7% unrecorded, χ^2^ 2.6, df 1, *p* = 0.11). However, female dogs were significantly more likely to be admitted for surgical sterilisation (53% vs. 47%, χ^2^ 16, df 1, *p* < 0.01). There was no significant change in the ratio of female to male dogs between ICCAHP years (χ^2^ 4.1, df 2, *p* = 0.13). A Kruskal-Wallis test showed there was no significant change in the age distribution of dogs presented to the ICCAHP between the first and subsequent years (*p* = 0.16). However, dogs presented to the ICCAHP that were already sterilised were significantly older than those that were entire (Figure 1b; *p* < 0.01). 

The proportion of dogs that were microchipped overall was low in the first year of the ICCAHP (27%) but varied significantly between communities (range 5.7–63%; χ^2^ 1210, df 12, *p* < 0.01). The proportion of dogs that were already microchipped overall increased significantly from 27% to 46% between the first and fourth year of the program (χ^2^ 15, df 1, *p* < 0.01). In Community 7 (the community with the longest period of involvement in the ICCAHP), the proportion of microchipped dogs increased consistently from 12% to 46% over four years (Figure 2). This represented a significant increase overall as well as significant increases between year 1 and 2 and year 3 and 4 (χ^2^ 28, df 1, *p* < 0.001; χ^2^ 5.9, df 1, *p* = 0.02; and χ^2^ = 6.7, df 1, *p* < 0.01 respectively).

Similarly, the proportion of dogs presenting to the ICCAHP that had already been surgically sterilised increased significantly between the first year of ICCAHPs (14%) to the fourth year (23%; χ^2^ 5.4, 1 df, *p* = 0.02). The increase was greater when only adult dogs (estimated age > 1 y) were considered: 23% were already sterilised in year 1 versus 37% in year 4 (χ^2^ 5.9, df 1, *p* = 0.02). The rate of surgical sterilisation amongst dogs presenting to the ICCAHP varied substantially between communities but increased in every community that participated for two or more years. Community 7 and Community 4 showed the most consistent increase; from 7.4% to 37.3% in Community 7 over four years (Figure 2; χ^2^ = 12, df 1, *p* < 0.001) and from 28.9% to 48.8% in Community 4 over three years (χ^2^ = 5.9, df 1, *p* = 0.015). 

Uptake of desexing by ICCAHP participants (i.e., the proportion of entire dogs that were surgically sterilised during the ICCAHP) varied between programs, ranging from 19% to 74%, with a mean of 48% in the first ICCAHP year. Uptake of surgical sterilisation was highest in most communities in the first visit year followed by a decrease in subsequent years. Likewise, for individual dogs, participants were more likely to opt for surgical sterilisation the first year of the ICCAHP than in subsequent years (uptake of sterilisation of 44% for first visits compared to 29% and 22% for second and third visits, respectively).

The proportion of dogs presenting to the ICCAHP that were return visitors from a previous year overall was 22% for both the second and third ICCAHP year. In Community 7, the proportion of dogs that had attended an ICCAHP in a previous year increased from 7% in the second year to 31% in the fourth year (Figure 2). 

### 3.2. Participant Feedback

Semi-structured interviews were conducted with 85 ICCAHP participants from nine communities. When asked the question “are there any good things about the dog program?” the majority of respondents (74 of 85, 87.1%) gave positive feedback about the ICCAHP and highlighted the need for the service in their communities: “helps people to look after their animals”, “keeps everybody happy, they look forward to it every year”. The main positive aspects noted were that the service was free (15 of 85 respondents, 17.6%) and that it was readily accessible in the community (17 of 85 respondents, 20%) “People don’t have the transport or the money to do this”, “convenient for the people to come here… otherwise people wouldn’t get their dogs treated.”, “lots of animals wouldn’t get vet treatment otherwise”. The importance of aiding with transporting animals was recognised as being an important component of the ICCAHP “helped me transport my dog from home to the [venue]”. The wider benefits of the ICCAHP to the community were also appreciated “helps reduce stray dogs on the streets”, “prevents strays and over-breeding. Makes it safer for kids”. 

Respondents frequently confirmed the importance of the relational aspect of the service delivery, highlighting their interactions with ICCAHP staff as a positive feature “great people, polite, courteous and friendly”, “people were nice”, “friendly people”. Comments from respondents also highlight the ICCAHP as an important opportunity to transfer knowledge “very informative”, “great advice”.

Only six of 85 respondents (7.0%) listed a negative aspect of the ICCAHP. In some instances, community engagement and promotion prior to the ICCAHP was problematic, meaning not everyone knew about the program ahead of time “only found out same day of the clinic, people were upset because they didn’t know about it”, “not much notice”, “not being advertised widely”.

When asked how the ICCAHP could be improved the main feedback concerned the need to provide an ongoing and more frequent service “should be two–three times a year”, “need more programs like it”, “it’s about time! Should be out here more often”. Others requested more education with the ICCAHP “attract more people in the community to learn things, such as pet health, control hygiene”, “involving schools, educating children about companion animals”, “educate the kids about kindness to animals”.

Respondents who had previously had a dog participate in the ICCAHP were asked if they noticed any changes in their dog following the program. Sixteen of 58 (27.6%) said ‘yes’, all of which were improvements. Respondents noticed a decrease in roaming behaviour “staying inside more”, “not roaming”, improved body condition and general health “they look healthier”, “shiny fur”, “put on weight”. Respondents also noted less nuisance behaviours in their dogs after the ICCAHP “not barking as much”, “they’re quieter and less messier [**sic**]”, “better behaved”. In our experience ICCAHP participants are often concerned that surgical sterilisation will change their dog’s personality; some respondents were pleasantly surprised “he hasn’t changed. If anything, he’s improved, looks a bit fatter”.

### 3.3. Secondary Source Indicators

Annual dog attack incidents for the whole of NSW and for regional NSW showed little variation between July 2012 and June 2019, remaining at approximately 0.6 and 1.0 dog attack incidents per 1000 population, respectively (Figure 3a,b).

Likewise, council pound statistics for the whole of NSW and for regional NSW showed little variation in total number of dogs admitted, number of dogs reclaimed by their owners and total number of dogs euthanased between July 2012 and June 2019 (Figure 4a,b). The decrease in numbers around May 2016 likely reflects changes to council pounds because of council amalgamations rather than a true decrease.

Data from LGAs participating for three or more years were pooled and analysed before and after commencing the ICCAHP. Pound data for total number of dogs euthanased per 1000 population did not meet the assumption of normality (W = 0.86, *p* < 0.01), and thus, was log-normal transformed prior to further analysis (W = 0.94, *p* = 0.25). LGA 6 was excluded from this analysis, despite participating for three consecutive years. LGA 6 was affected by council amalgamations in 2016 and had missing data for several years. In addition, the ICCAHP in Community 1 (the participating Community in LGA6) did not include surgical sterilisation until its third year.

Based on pooled data from LGAs 1, 2 and 8, dog attack incidents per 1000 population were significantly lower following commencement of the ICCAHP than before the intervention (F(2,1) = 5.4, *p* = 0.035). All three LGAs reported annual dog attack incidence of less than 1 per 1000 population for the 2018–2019 financial year, an incidence comparable with the average for regional NSW (Figure 5).

The number of dogs euthanased in council pounds was also significantly lower following commencement of the ICCAHP in the pooled analysis (F(2,1) = 14, *p* < 0.01), with LGAs 2 and 8 achieving euthanasia per 1000 population comparable to the average for regional NSW (Figure 6). This represents a 98% reduction in LGA 2, 79% in LGA 1 and 65% in LGA 8. The reduction in euthanasia coincided with significantly more dogs being released by these council pounds to organisations for rehoming (F(2,1) = 14, *p* < 0.01), which in LGA 1 and LGA 2 started from close to zero before the ICCAHP to 80% of total dog admissions for the 2018–2019 financial year. LGA 5, home to a community that received a single ICCAHP (hence, it was not included in the pooled analysis), also reported a 70% reduction in the number of dogs euthanased at the council pound following its first ICCAHP in 2018. Reporting of the reasons for euthanasia were inconsistent between LGAs; hence, these were not analysed separately.

There was no significant change in the total number of dogs admitted to council pounds in the pooled analysis (F(2,1) = 0.73, *p* = 0.41); however, individual LGAs did record large reductions. In LGA 2, for example, the total number of dogs admitted to the council pound has steadily decreased from 80 dogs per 1000 population (amongst the highest recorded intakes in the state) in 2015–2016, the year prior to the ICCAHP commencing, to 25 dogs per 1000 population in 2018–2019, a 68% reduction. Likewise, in LGA5, the total number of dogs admitted to the council pound was 56% lower the year following their first ICCAHP compared to the year prior.

Community 2 in LGA 7 presents an important contrast. This community received its first ICCAHP in November 2019, after the most recent council data; hence, this community presents a natural no-intervention control. In LGA 7, the total number of dogs admitted to the council pound per 1000 population has continued to increase by an average of 2.8 dogs per 1000 population every year, and in 2018–2019 was more than six times higher than the average for regional NSW (Figure 7a). Dog attack incidents in this LGA have also consistently been greater than 4 per 1000 population since 2015—more than four times the average for regional NSW—and peaked at 8.1 per 1000 population in the 2018–2019 financial year (Figure 7b).

## 4. Discussion

Continuous monitoring and evaluation of any intervention is necessary to determine its effectiveness, to inform improvements to program delivery and to provide evidence for continuation of the program [24]. This study demonstrates that, despite their limitations, routinely collected data can be used to monitor and evaluate DPM interventions, their major advantage being accessibility and low cost. We also demonstrate the potential for such interventions to significantly reduce dog attack incidents and the number of dogs being euthanased in council pounds. Our findings reinforce previous recommendations that DPM interventions prioritise engaging communities with dog health activities to maximise impacts for community health and safety and for dog welfare.

The ICCAHP was associated with significant reductions in dog attack incidence, an important indicator of community safety. This is a key indicator, because it describes both impacts on public health risk and risks to livestock and wildlife by including attacks and threatening behaviour towards people and other animals. These are important impacts to participating communities as well as to other key intervention stakeholders, including local government, livestock producers and human health services. Dog bite injuries are an important cause of morbidity, especially in children, resulting in high rates of hospital admission and considerable psychological trauma even without the threat of rabies [1,25]. Menacing free-roaming dogs also affect the liveability of a community by restricting the freedom of people to move around their neighbourhoods and preventing physical activity [10]. Dog population management interventions in India have previously been associated with a reduction in human rabies incidence, most cases which result from dog bites [16]; however, evidence of impacts on community safety in Australia are lacking [13]. 

We consider the reduction in dog attack incidence to be due to a combination of a reduction in the roaming dog population and an increased proportion of sterilised dogs. The behavioural impacts of sterilisation reported in the literature differ depending on whether dogs are considered on an individual or a population basis. Sterilisation of individual dogs, both male and female, has been associated with increased fear and aggression, both towards familiar people, unfamiliar people and other dogs [26,27,28]. However, entire dogs are more often implicated in dog bites on humans than sterilised dogs [29,30]. Sterilised dogs have also been shown to have smaller home ranges than entire dogs, suggesting a reduced tendency to roam [31]. Sterilisation is also associated with a decreased risk of various health conditions, decreased risk of injuries and increased life expectancy, aside from the obvious benefit of permanent reproductive control [28,29].

The ICCAHP was also associated with a significant reduction in rates of euthanasia of dogs in council pounds, some of which had amongst the highest rates of euthanasia in NSW. In some LGAs, this coincided with a reduction of total admissions of dogs and in others to increased release of dogs to other organisations for rehoming. While this change cannot be directly attributed to the intervention, partnerships with councils resulting in increased awareness of alternative rehoming options, and increased community awareness of dog welfare resulting from participation in ICCAHPs potentially contributed. 

Rates of euthanasia are a key indicator for animal welfare impacts as well as reflecting perceptions of dogs and their value in a community [6]. There is strong existing evidence of the animal welfare benefits of DPM interventions. Studies of large, long-running Animal Birth Control (ABC) programs in India have demonstrated that dogs previously sterilised through the programs have significantly superior health compared to sexually entire dogs, attributed to behavioural changes and reduced energy requirements of sterilised compared to entire dogs [16]. In addition, studies have demonstrated that entire (untreated) dogs in cities with regular ABC programs were significantly healthier than their counterparts in cities without such interventions, having higher body condition scores, lower prevalence of open wounds and lower prevalence of fleas [17]. Smaller studies from Australia support these findings [32,33]. Although indicators on the perception of dogs and attitudes towards dogs can be more difficult to measure, these are valuable to determine if an intervention is changing social norms.

Study findings illustrate how performance on various indicators relate to community engagement. The uptake of surgical sterilisation by participants—an important factor in the success of any DPM intervention [6,13]—was high the first year, but fell substantially in subsequent years, and this finding was consistent across the participating communities. This highlights two different challenges to DPM that require different approaches. The first challenge is a dog health service gap and a lack of accessibility of veterinary services, i.e., dog owners who value veterinary services (e.g., surgical sterilisation) but are unable to access them due to barriers such as availability and cost. This challenge is relatively easy to address by provision of accessible and affordable services through free or subsidised programs. The second challenge is a lack of engagement with dog health activities, i.e., dog owners who may love and value their dog but do not see value in veterinary services such as surgical sterilisation and will not actively engage with these services even when they are free [12,25,33]. This second challenge, if not effectively addressed, will necessarily limit the impact of a DPM intervention.

Knowledge exchange and providing access to culturally inclusive dog health information is important to improve community engagement with dog health activities and this has been reported extensively [6,34,35]. How a program is delivered is arguably more important. A focus on building trust between service providers and communities and empowering communities to own the solutions to their DPM needs is likely to have the greatest long-term impacts on changing norms around dog welfare and management practices [12,25,36]. In addition, as was highlighted by the responses from participants, reaching out directly (i.e., going door-to-door) to individuals who might not realise they have anything to gain from participating in a program is likely to be necessary in communities where a lack of engagement with dog health activities is a major challenge to DPM. 

We propose quantifiable indicators of community engagement that can be used to evaluate intervention performance. Uptake of services such as surgical sterilisation and number of dogs presented or sterilised per unit population, for example, are indicators that are easily monitored through program metrics. Impacts on secondary indicators such as dog attack incidence are likely to be mediated by many factors beyond the number of dogs sterilised, for example, through knowledge exchange with program participants and improved relationships between communities and other stakeholders such as local government dog control officers and local veterinarians. Nonetheless, the findings of this study suggest communities with greater engagement (more dogs presented, higher uptake of services) demonstrated greater reductions in dog attack incidents, council pound euthanasia and pound admissions. 

To maximise impacts of a dog health program, based on the findings of this study, we suggest aiming to: (1)Have at least 100 dogs per 1000 population of the target community present to the program.(2)Surgically sterilise 50 dogs per 1000 population.(3)Have an uptake of surgical (or chemical) sterilisation of 50% or greater.

Routine measurement of these indicators allows organisers to identify when programs are under-performing. Underlying barriers and drivers of engagement can then be explored and interventions refined when needed. For example, if a low uptake of sterilisation is found to be due to fears around anaesthetic complications, or concerns about impacts on masculinity, these concerns can be addressed directly through targeted awareness and education campaigns.

This study also revealed that the commencement of an ICCAHP in a community frequently coincided with a peak in indicators such dog attack incidence and total council pound admission. This reflects the current model of practice in NSW, where DPM is often instituted in the short term in response to an immediate dog-related crisis. This highlights the potential benefits of a shift towards sustained, long-term investment in providing targeted free or subsidised dog health services [6].

This study has limitations, which are illustrative of the practical challenges associated with effectively evaluating the impact of a dog health intervention. The current impact evaluation was heavily dependent on the quality of the data collected by local government, which varied considerably between jurisdictions. Several council datasets had missing values and data were only available at the LGA level, which limits the resolution of the analysis. This highlights the value of councils keeping accurate records of dog complaints and council pound admissions and the value of this data being publicly available. Ideally, this would include the collection of data on the sex and neuter status on dogs admitted and euthanased at municipal animal shelters and the reasons for euthanasia, and the sex and neuter status of dogs involved in dog attacks. In addition, because impacts are measured indirectly and because selected indicators are influenced by multiple factors beyond the intervention investigated, it is impossible to directly attribute changes in measures to the ICCAHP. Factors such as council staff turnover or changes in council pound operating hours, for example, could have important effects on these indicators, especially in smaller LGAs. Furthermore, it is not possible to rule out regression to the mean as an explanation for some of these findings. We have attempted to overcome this limitation through replication in multiple communities and by using diverse indicators.

Another limitation is the lack of data on the overall dog population in participating communities. It is unclear how representative the participating population is of the target population. Communities included in this study are not well suited to mark-resight methodologies as recommended elsewhere [15] because participating communities are often small, and the numbers of free-roaming dogs are relatively low and constitute only a small proportion of the total dog population—the majority being owned and confined to private property [12]. This is likely to apply in most towns and cities in Australia and other developed countries. A dog census amongst the target population for the intervention would be more appropriate [35]. As well as a count of the number of dogs, a census allows an estimation of the reproductive rate (through counting the number of lactating females and juveniles), rates of sterilisation, ratio of females to males and assessment of animal welfare indicators such as body condition score, skin condition and the prevalence of injuries or specific conditions such as transmissible venereal tumours [6]. A census can also be used as a baseline against which program metrics can be measured such as uptake of veterinary services (e.g., health checks and surgical sterilisation) as an indicator of community engagement. However, conducting a census can be resource-intensive. Our recommendation to maximise its cost effectiveness is to prioritise a census before commencing a DPM intervention, which can then be repeated at intervals, if funding allows. A quarterly or annual census would be ideal in situations where this is practical; however, a census every five years is likely to be sufficient for effective monitoring and evaluation in most circumstances. Repeating a census intermittently allows progress towards additional impacts—such as changes to dog population density and dog population turnover—to be evaluated [6].

## 5. Conclusions

Dog population management interventions can have important impacts on improving community safety, improving dog welfare and increasing community engagement with dog health activities. Multiple readily measured indicators can be used in the monitoring and evaluation of DPM programs, many of which are collected by local government. Their utility can be maximised by carefully selecting indicators that reflect the objectives of each specific program, which will vary between communities and target dog populations. Community engagement with dog health activities is arguably the most important impact affecting all other outcomes of a DPM intervention. We propose indicators to measure community engagement and suggest benchmarks to guide program planning and evaluation. Further studies to validate these and other potentially valuable indicators are warranted.

## Figures and Tables

**Figure 1 animals-10-01061-f001:**
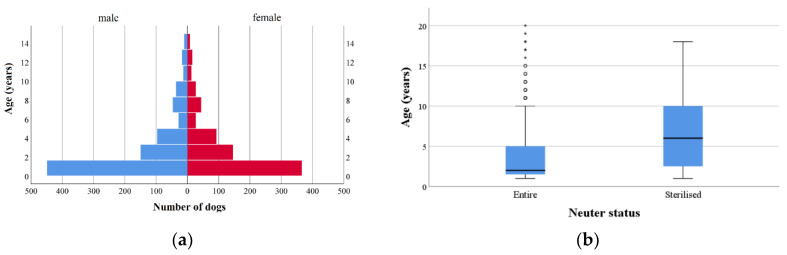
Age distribution of dogs presented to the RSPCA Indigenous Community Companion Animal Health Programs in New South Wales, Australia between September 2016 and November 2019: (**a**) overall; (**b**) by neuter status.

**Figure 2 animals-10-01061-f002:**
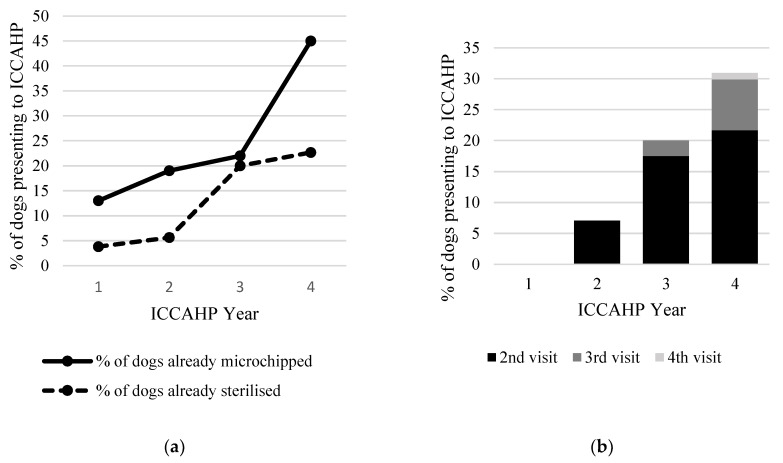
(**a**) Proportion of dogs presenting to an Indigenous Community Companion Animal Health Program (ICCAHP) in NSW, Australia between September 2016 and November 2019 that had previously been microchipped (solid line) and were already surgically sterilised (broken line) over four consecutive years in Community 7. (**b**) The proportion of dogs presented to the ICCAHP in Community 7 that were attending the ICCAHP for a second, third and fourth year.

**Figure 3 animals-10-01061-f003:**
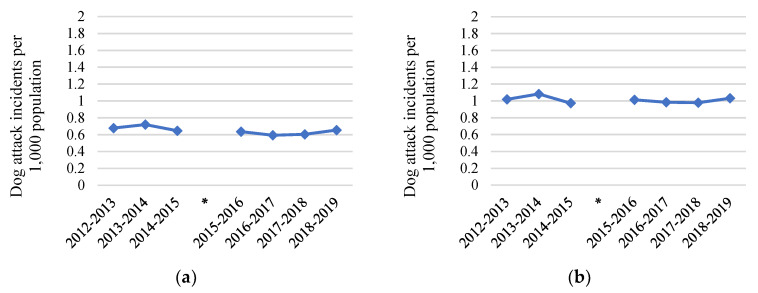
Total dog attack incidents per 1000 population for: (**a**) the whole of New South Wales (NSW); (**b**) the whole of regional NSW. The * indicates timing of council amalgamations that affected 39 of 152 LGAs in NSW, including one LGA containing a community included in the study (LGA 6).

**Figure 4 animals-10-01061-f004:**
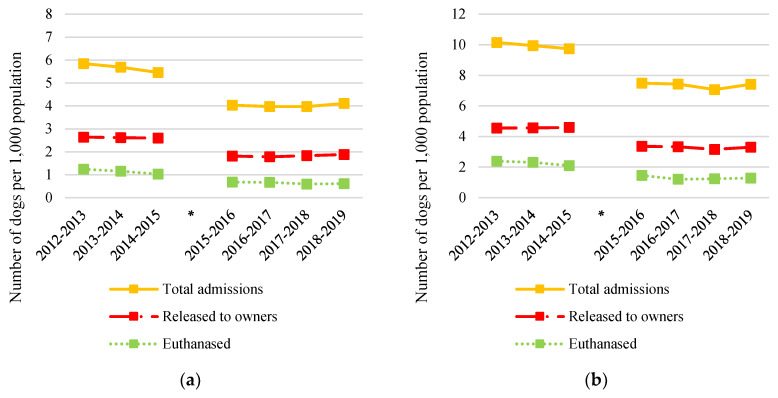
Council pound statistics on total number of dogs admitted, number of dogs released to their owners and total number of dogs euthanased per 1000 population for: (**a**) the whole of New South Wales (NSW); (**b**) for the whole of regional NSW. The * indicates timing of council amalgamations which affected 39 of 152 LGAs in NSW, including one LGA containing a community included in the study (LGA 6).

**Figure 5 animals-10-01061-f005:**
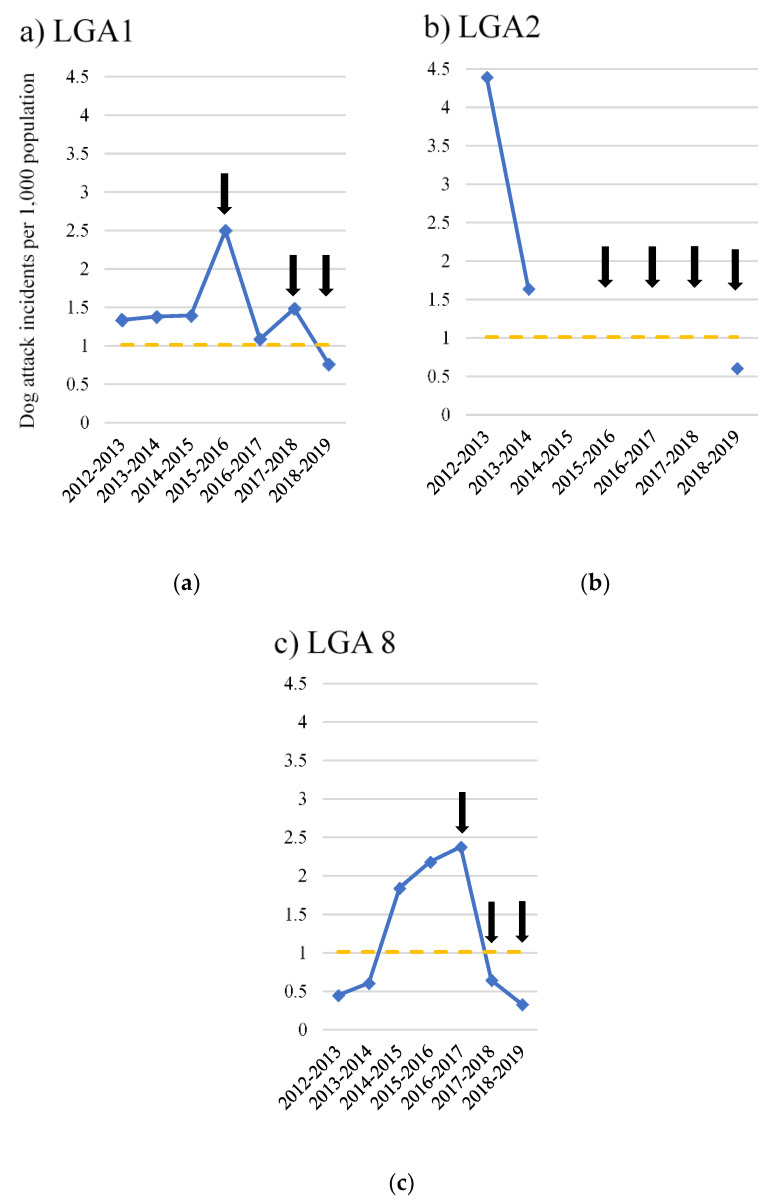
Total dog attack incidents per 1000 population for Local Government Areas (LGAs) participating in a dog health intervention, the RSPCA Indigenous Community Companion Animal Health Program (ICCAHP), for three or more years: (**a**) LGA 1; (**b**) LGA 2; (**c**) LGA 8. Black arrows indicate the timing of ICCAHPs. The broken yellow line indicates the average annual dog attack incidence for regional NSW (1.0 per 1000 population). N.B. Data between 2014–2015 and 2017–2018 was not available for LGA 2

**Figure 6 animals-10-01061-f006:**
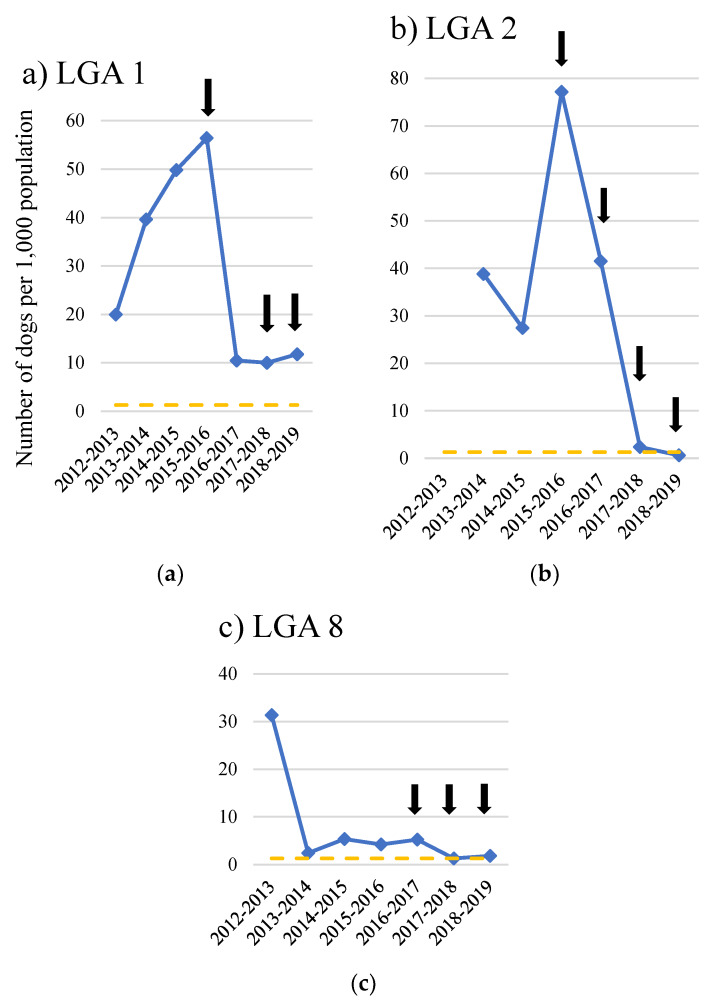
Number of dogs euthanased by council pounds per 1000 population for Local Government Areas (LGA) participating in a dog health intervention, the RSPCA Indigenous Community Companion Animal Health Program (ICCAHP), for three or more years: (**a**) LGA 1; (**b**) LGA 2; (**c**) LGA 8. Black arrows indicate the timing of ICCAHPs. The broken yellow line indicates the average number of dogs euthanased for regional NSW (1.3 per 1000 population).

**Figure 7 animals-10-01061-f007:**
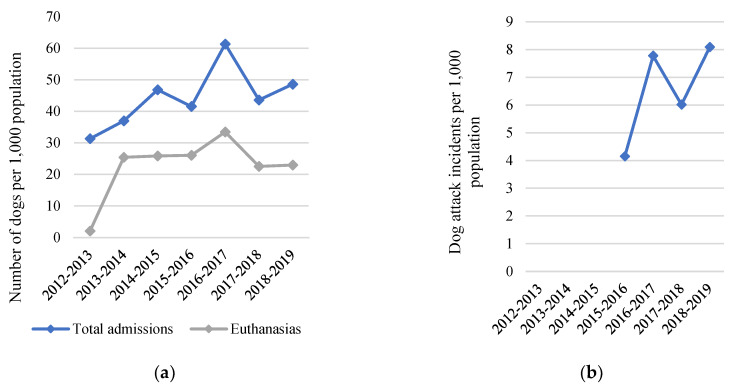
(**a**) Total number of dogs admitted, and euthanased at the council pound per 1000 population; and (**b**) number of dog attack incidents per 1000 population for one Local Government Area (LGA 7) until the year prior to participating in a dog health intervention, the RSPCA Indigenous Community Companion Animal Health Program (ICCAHP). N.B. Data on dog attack incidents was not available before 2015–2016.

**Table 1 animals-10-01061-t001:** Indigenous Community Companion Animal Health Programs (ICCAHPs) Conducted by the Royal Society for the Prevention of Cruelty to Animals (RSPCA) in New South Wales, Australia between September 2016 and November 2019.

Community (LGA)	Remoteness ^1^	Access to Veterinarian Locally	Target Population for ICCAHP ^2^	Target Population as Percentage of the LGA Population	2016	2017	2018	2019	Dogs Presented to ICCAHP (% Entire)	Dogs Sterilised (Uptake of Sterilisation ^3^)	Dogs Presented per 1000 Target Population	Dogs Sterilised per 1000 Target Population
7 (2)	Very remote	No	1000–2000	69.1	**X**				105 (96)	58 (57.4)	91.9	50.7
						**X**			71 (94)	45 (67.2)	62.1	39.4
							**X**		80 (80)	21 (32.8)	70.0	18.4
								**X**	97 (77)	37 (49.3)	84.9	32.4
6 (1)	Remote	Yes	1000–2000	69.4	**X**				115 (91)	53 (50.5)	63.0	29.1
							**X**		75 (79)	11 (18.6)	41.1	6.0
								**X**	89 (82)	30 (41.1)	48.8	16.4
1 (6)	Inner regional	Yes	10,000–15,000	15.8		**X**			65 (82)	N/A	4.0	N/A
							**X**		48 (85)	N/A	2.9	N/A
								**X**	68 (81)	13 (23.6)	4.2	0.8
4 (8)	Remote	No	100–500	7.2		**X**			68 (84)	35 (61.4)	156.3	80.5
							**X**		34 (68)	6 (26.1)	78.2	13.8
								**X**	57 (65)	14 (37.8)	131.0	32.2
11 (2)	Very remote	No	100–500	14.9	**X**				34 (100)	15 (44.1)	137.7	60.7
						**X**			16 (38)	4 (66.7)	64.8	16.2
12 (2)	Very remote	No	<100	4.7	**X**				35 (97)	13 (38.2)	454.5	168.8
						**X**			35 (94)	13 (39.4)	454.5	168.8
10 (1)	Very remote	No	100–500	5.6	**X**				35 (94)	21 (63.6)	236.5	141.9
							**X**		19 (74)	3 (21.4)	128.4	20.3
8 (3)	Very remote	No	500–1000	40.6			**X**		98 (61)	21 (35.0)	131.5	28.2
								**X**	73 (74)	28 (51.9)	98.0	37.6
3 (5)	Outer regional	No	500–1000	3.8			**X**		75 (89)	48 (71.6)	128.2	82.1
5 (8)	Remote	Yes	2000–5000	35.4				**X**	76 (93)	42 (59.2)	35.4	19.6
9 (3)	Very remote	No	100–500	10.7				**X**	26 (73)	12 (63.2)	132.7	61.2
13 (4)	Inner regional	Yes	10,000–15,000	49.7				**X**	71 (85)	26 (43.3)	4.8	1.8
2 (7)	Outer regional	No	500–1000	5.6				**X**	68 (68)	12 (26.1)	90.3	35.9
								**Min**	16 (38)	3 (18.6)	2.9	0.8
								**Max**	115 (100)	58 (71.6)	454.5	168.8
								**Median**	68	21 (43.7)	87.6	32.3

^1^ According to the Australian Statistical Geography Standard [21]; ^2^ Estimated size of the human population eligible to participate in the ICCAHP; ^3^ Percentage of entire dogs that presented to the ICCAHP that were surgically sterilised during the ICCAHP; LGA = Local Government Area.

**Table 2 animals-10-01061-t002:** Indicators used to evaluate impacts of a dog population management intervention in NSW, Australia, 2016–2019 (adapted from ICAM [6]).

Impact	Indicator
Dog population turnover	Age structure % of dog participants from previous years returning to the ICCAHP Total council pound admissions
Community Engagement	% microchipped % sterilised Dogs presented per 1000 population Dogs sterilised per 1000 population Uptake of sterilisation Participant feedback
Community Safety	Dog attack incidents per 1000 population
Dog welfare	Dogs euthanased in council pounds per 1000 population Dogs released by council pounds to rehoming organisations Female to male ratio

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
