# Peer review of "Evaluation of a Dog Population Management Intervention: Measuring Indicators of Impact"

_animals, 2020, doi:10.3390/ani10061061_

Round 1
Reviewer 1 Report
Thank you for the opportunity to review this paper. The authors evaluated effectiveness of dog population management interventions conducted in years 2016-2019 in New South Wales (Australia). Especially for Aboriginal community identified need for a dog health intervention providing by the Indigenous Community Companion Animal Health Program. Participants accessed free veterinary consultations, vaccination, permanent identification and sterilization. The authors prove that in a result of intervention within the ICCAHP significant reductions in dog attack incidence occurred which is very important for public health risk. A significant reduction in rates of euthanasia of dogs in council pounds were also noticed which is evidence of an increase in dog welfare. This is an interesting and soundness paper and I think it would be of interest to readers of Animals. The manuscript is well written, however, I feel that it needs some revisions before it should be accepted for publication.
There are some weaknesses that the Authors would need to address and build on the paper, in particular, sex ratio of examine dogs in some aspects. According to data presented in the submitted manuscript female dogs were significantly more likely to be admitted for surgical sterilisation than males. The authors present as secondary source indicators associated with positive result of the ICCAHP - a reduction in the number of dog attack incidence and a significant reduction in rates of euthanasia of dogs in council pounds. Such conclusions would be justified by providing the following data and correlating them with each other: sex ratio of attacking dogs, sex ratio of dogs admitted council pounds and their neuter status, sex ratio of dogs euthanized in council pounds and their neuter status. Also, what are the reasons for euthanasia in council pounds? Why the number of euthanasia in council pounds has decreased, since according to the authors the number of dogs admitted has remained unchanged.
I also think that the issue of the effect of desexing on dog behaviour should be discussed. In the light of current knowledge, the answer to this question is not so obvious.
Minor revisions
Line 268 – I can’t find this “2” in Table 2
Line 319 – It should be “Figure 4a and b” instead of “Figure 3a and b”
For me presentation of data on Figure 2 is unclear.
Also, I think it would be more clear to use the term neutering or desexing instead of sterilization/sterilisation.
Reviewer 2 Report
Just a few comments and/or change suggestions
It might be helpful to provide a bit more background on the communities attitudes towards owned and unowned animals. Are they valued as companions or as working dogs? What is the general attitude towards providing Vet care? What is the percentage of the community that owns animals? Forgive my lack of knowledge here but isn’t there a type of dog - dingo - that is concerned wild? What role doesn’t that play in this study? Did this intervention include community education? So perhaps a more focused section would n the community might add additional depth to the study
Methods Question
you stated you did personal interviews who did the interviews, was consent obtained and how were human subjects protected
how many interviews and did you collect any data on responses by subgroups
Another question:
Does potential negative attitudes towards neuter or spay play into push back against these services? For instance breeding for working dogs or worries about masculinity impact sterilization rates?
Overall an excellent research study that evaluates a program that addresses a serious social problem. I found it very interesting as you can tell from my questions.
Reviewer 3 Report
This is a nicely written up project on some potentially useful and simple parameters to measure success and impact of dog population control efforts. Unless there is a compelling reason to keep the term “pound” most countries have moved to using animal shelter. For shelters which are funded by government, they are usually referred to as municipal, government-funded or public shelters. Please edit throughout.
I have some specific comments below.
Line 25: please add that this was across at least 3 years. And please also note or edit that the decrease in euthanasia corresponded to an increase in transfers out of the council shelters.
Line 94: the first 2 sentences of the methods are really introduction. Please move to line 85.
Line 105: this “relative socioeconomic disadvantage” is a bit cryptic for an international audience. Is there some cut off or percentile or other definition that can be included in the text to make this more clear?
Line 108: tables should be numbered in the order they are found in the text.
Table 1: Indicator for euthanasia; I think it would be clearer and stand alone to state that this is in the shelters. For “rehoming to organization” typically this is termed surrender or relinquishment. The shelter isn’t a new home, a new owner is and that is an important transition for animal welfare organizations.
2.3.2: were there any reporting or tracking changes? I would include the shift in transfers out here as well. Anything that would influence the data over the time period of interest that would also help explain the findings. And please emphasize in the discussion that attempting to identify these sorts of changes, including staffing decreases or more limited hours of operation of the shelters, is critical because these can influence the indicators profoundly even in the absence of an intervention. So, they are important to consider as potential sources of confounding or bias.
Line 196: please clarify which variables were examined for statistical associations here? It seems a bit haphazard in the results (section 3.1).
Data starting line 231: please add the microchip % data to Table 2. That will make the patterns easier to find.
Line 234: I believe that this is the overall % microchipped in all communities as opposed to the range of % microchipped for all communities? Please edit in the text and clarify.
Line 252 has a typo in “yea-+r” at least in my copy ?. These results would correspond to the idea of low hanging and high hanging fruit: the people who were really interested and easy to engage participated early on and the people who were less engaged or more reluctant to have a dog sterilized will likely need more and longer efforts to have them to the point of getting their dog desexed.
Table 2: I don’t see where footnote 2 appears in the table. Without the text, the differences between inner and outer regions are opaque. Please clarify, perhaps in a footnote. Also order by remoteness and then by some other variable that is important, so patterns are easier to see. In the dogs presented to ICCAHP (no. entire) I think that the % entire would be a lot easier for the reader to take in and see who has a lot and who doesn’t. For dogs sterilized (uptake of sterilization) I think that uptake is the % of entire dogs presented? That isn’t clear from the table so please edit. Also note that the summary stats (at least in my copy) are mis-aligned by one column.
Paragraph starting line 291: This is interesting feedback given the limited uptake. Does it seem to be reasonable as to why more people didn’t come?
Line 302-4: was this statement that “respondents were often concerned that surgical sterilization…” from the open ended discussion as well? Or from preexisting knowledge? Please clarify.
Lines 314 and 320: please clarify “council amalgamations” and the implications for those for the shelters in the methods with the changes during the study period. Again, this is an international audience and the implications may not be completely clear.
Lines 332-4: these are from the methods, not the results. Please edit and include the results from these tests.
Figure 3: why is there missing data in LGA2? Please add to the figure and text. Could these results also be due to regression to the mean? Please consider in the discussion.
Line 358: yes, and this likely means that changes in euthanasia can’t be ascribed to the intervention. Please edit throughout.
Paragraph starting line 382: This sort of contrast seems to be fairly common. Please be sure to add to discussion and share any reasons why this may be. It is possible that the ICCAHP increased awareness of the shelter and reporting and or that the intervention isn’t big or long enough to have impact. Or that there are other factors in this community driving these rates. Makes one wonder if the other communities’ data would have changed without the intervention…something we don’t know with community level work and ecological analyses.
Line 412: these first 2 sentences don’t seem to connect to the rest of the paragraph. Euthanasia in shelters isn’t necessarily due to health or behavior; it can be due to resources, space, etc. Please edit this paragraph.
Starting line 426 and the rest of the page: Very good discussion of issues. How were the total number of dogs established so that % presented could be calculated?
Line 459-61: what are these based on? Please be clear in the discussion.
Line 479: I think that another limitation is that these simpler measures haven’t been validated against more intensive ones. So, they are kind of proxies where the extent and direction of bias isn’t known (yet).
